# A Sustainable Iterative Product Design Method Based on Considering User Needs from Online Reviews

**Qi Wang** [1], **Shuo Wang** [1] **and Si Fu** [2,*]

1   School of Industrial Design, Hubei University of Technology, Wuhan 430068, China;
    wq20201103@hbut.edu.cn (Q.W.); 102111408@hbut.edu.cn (S.W.)
2   China-Korea Institute of New Media, Zhongnan University of Economics and Law, Wuhan 430068, China
*   Correspondence: z0004435@zuel.edu.cn; Tel.: +86-159-0272-0781

**Abstract:** Small and medium-sized manufacturing industries can use online reviews to add valuable user requirements, enabling them to iteratively and precisely upgrade their products based on user needs. However, a sustainable, iterative approach to product design requires the integration of a large amount of information about user requirements for accurate selection. Currently, product iterations are primarily focused on developing new solutions or upgrading a few components with little screening to see if the product iterations meet user needs. This leads to a large number of wasted resources and a shortened product lifecycle. To address these challenges, this paper proposes a sustainable iterative research method that mines user needs and provides comprehensive decision making for product design based on online reviews, using probabilistic semantic term sets (PLTS). The proposed method considers the hesitation and uncertainty among evaluating experts regarding indicators, and uses the decision-making trial and evaluation laboratory (DEMATEL) method to analyze the correlations between demand indicators. The DEMATEL correlation function is improved by reconstructing the PLTS acquisition score function and deviance into a DEMATEL correlation function, in the form of exact values using an improved binary semantic approach. This iterative design approach provides accurate feedback on how users feel about the use of product components and ensures that most product components are sustainably recycled. A drone case study is presented to demonstrate the feasibility of this approach. In-depth interviews with experts confirm that this approach is more sustainable and provides a new research methodology for sustainable iterative product design.

**Keywords:** user needs; probabilistic semantic sets; iterative product research; multi-attribute decision methods; sustainability

## 1. Introduction

The sustainability of Internet and Industry 4.0 development relies heavily on the manufacturing industry, particularly in terms of assessing its rapid development from an environmental perspective and rationing the use of limited global resources while safeguarding environmental sustainability [1]. Small and medium-sized manufacturing industries use most of the world's resources, and studies have shown that natural resources are negatively correlated with a country's ecological and economic growth, leading to the over-consumption of resources, water and air pollution, and the production of significant amounts of highly polluting waste [2]. The large-scale development of green products in small and medium-sized manufacturing industries has resulted in the rapid obsolescence of older, non-green products. This shortens the life cycle of existing products and generates a significant amount of waste [3].

As a result, there is widespread interest in sustainable products versus existing non-sustainable products, and small and medium-sized manufacturing industries need new ways to improve sustainability and slowly replace non-sustainable products [4]. Currently,

the most cited sustainability theory for reducing the environmental impact of production and consumption is the Product Service System (PSS), proposed by Mont [5]. Feng et al. [6] proposed treating existing product modules as original systems and combined the Analytical Hierarchy Process (AHP), Lean Design-for-X (LDfX), Design Structure Matrix (DSM), and Pearson Correlation Coefficient (PCC). However, small and medium-sized manufacturing industries may require a new service system based on PSS theory to address sustainable product iteration solutions, which require significant resources. Furthermore, most small and medium-sized manufacturing industries are unable to solve this problem by themselves, not only due to their limited resources and service capabilities, but also because overly complex solutions increase their costs and risks [7].

The multi-attribute decision problem plays a crucial role in sustainable iterative product design, and previous work proposed an iterative conceptual design process (S-KFCF) that integrates the KANO model, FBS model, and FAHP method [8]. This paper proposes a new approach that leverages online reviews to identify user requirements and precisely corresponds to product structure modules. However, the real-time nature of online reviews and their continuous updating makes mining and filtering user needs a significant challenge. To address this issue, this paper proposes a sustainable iterative design methodology that considers the limited resources and service capabilities of small and medium-sized manufacturing industries. This approach extends product lifecycles, reduces waste, and reduces costs and risks for these industries.

This paper is organized as follows: Section 3 outlines the research methodology, mathematical algorithms, and models; Section 4 provides case experiments and validation of the results; and finally, Section 5 presents conclusions and future research directions.

## 2. Related Work

Small and medium-sized manufacturers can effectively use online reviews to make decisions and gain a competitive advantage [9]. In previous studies, it has been observed that decision making plays a crucial role in the various aspects of life. In product iteration, decision making is the fundamental process that leads to different directions of product development [10]. Yang et al. [11] found that traditional user research methods faced decision-making challenges in the process of product design and improvement. They proposed a new method to obtain useful online reviews from e-commerce platforms, constructed a product evaluation index system, and proposed product improvement strategies for online reviews. Yu et al. [12] believed that decision making occurs at all stages of product design, and proposed using multi-index decision making to improve the continuity and efficiency of product design and to solve problems in the product design process. Multi-attribute decision making (MADM) is a widely used method to select an optimal choice from a limited number of alternatives based on multiple attributes [13,14]. Thus, introducing MADM can effectively address the problem of decision making in the product iteration process. In MADM, it is essential to present information objectively under each attribute, and semantic terms are employed to represent significant information about the model, which can include evaluation information or qualitative assessment information [15,16].

In the face of rapid economic development, decision makers increasingly use a variety of semantic terms to provide evaluation information about product characteristics. Rodriguez et al. [17] proposed a hesitant fuzzy semantic term set (HFLTS) to represent the multiple semantic terms of a participant. Wang [18] further developed the extended hesitant fuzzy semantic term set (EHFLTS), which merges with HFLTS. The probability distribution of each semantic term is equal in either HFLTS or EHFLTS, but different participants may choose different semantic terms. To describe the level of preference for these semantic terms, Chen et al. [19] introduced a set of proportionally hesitant fuzzy semantic terms consisting of semantic terms and their corresponding probability distributions. Currently, participants may provide incomplete partial probability distributions or may ignore them. Pang et al. [20] proposed probabilistic linguistic term sets (PLTS) to define incomplete probability distributions, which can be considered a general form of PHFLTS, HFLTS, and



EHFLTS. PLTS assigns different probabilities to the semantic terms to represent different degrees of preference, thereby precisely translating the hesitation and uncertainty present in the verbal evaluation of experts. PLTS is more flexible and comprehensive than other models and can appropriately describe complete and incomplete assessment information. Qualitative information is effectively addressed in the decision-making process and computed using expressions [21]. While PLTS can address hesitancy and uncertainty in expert scoring, it does not allow for the analysis of correlations between indicators. PLTS has been used to conduct several studies on probabilistic semantic environments, including the PLTS weight determination model [22,23] and probabilistic semantic decision-making methods. However, while PLTS can address the hesitancy and uncertainty of verbal evaluations in expert scoring, it does not allow for the analysis of correlations between indicators. For example, in a given product, the product components that users are concerned about are the same, but because of the limited range of user reviews there will be subtly different review expressions for the product, yet most real user review information is vague, uncertain, and inaccurate [24]. Therefore, it will be difficult for the model to identify a large amount of duplicate review information.

To address this problem, Baykasoğlu et al. [25] proposed the decision-making trial and evaluation laboratory (DEMATEL) to consider the interdependencies between problem attributes. Yi et al. [26] reconstructed PLTS elements and developed a multi-attribute decision approach based on the DEMATEL method, adding probabilistic information and extending the DEMATEL method to the re-customized PLTS framework. The DEMATEL method calculates deterministic values when computing the normalized and total correlation matrices between the indicators. Therefore, this paper proposes a new method to consider the hesitation and uncertainty of experts when evaluating the information extracted from user requirements. We first calculate the score functions associated with the indicators and then transform them into exact values using binary semantics [27,28]. This paper aims to use multi-attribute decision making to make optimal sustainable product iterative design solutions that meet user needs, reduce the cost and risk of iteration for small and medium-sized manufacturing industries, and improve the ability to evaluate and improve product iterations. The proposed method will allow for the analysis of correlations between indicators, addressing the issue of duplicate review information in real user review data, and enhance the ability to identify product components about which users are concerned.

## 3. A Sustainable, Iterative Approach to Product Design

This study employs online reviews to identify user needs and proposes a multi-attribute decision-making approach to address the sustainable product iteration problem. The approach presented in this paper explores how small and medium-sized manufacturing industries can use online review information to capture user needs and iterate products in a sustainable manner.

The sustainable iterative approach to product design proposed in this study involves the following steps: (1) Obtaining user needs information from a large number of online reviews. (2) Thematically organizing user requirements using the BTM model to establish a product iteration factor information base. (3) Dividing the corresponding products into structural modules to align user requirements with the structural modules as a second screening for user requirements. (4) Introducing the PLTS method to account for expert scoring hesitation and uncertainty. The DEMATEL method is used to consider the interrelatedness of indicators and is reconstructed into a DEMATEL correlation function in the form of exact values using an improved binary semantic approach. This step is a core element of the iterative design. (5) Analyzing the correlation rankings of the importance of the structural modules of interest to the users in order to prioritize the modules for the final iteration. (6) Discussing a sustainable product iteration plan in focus groups with professional product designers, structural engineers, brand owners, and product development technicians. The discussion focuses on the selected critical structural and technical modules; the functional characteristics of the components, user requirements, and

the current state of the product are gradually analyzed to ensure the sustainable recycling of most modules and the iterative upgrading of the product.

The entire process of sustainable iterative product design involves an iterative filtered loop, where each horizontal plane clearly depicts the iteration stage, and each stage can be paused, as shown in Figure 1. This iterative process can be repeated with the original product to meet changing user needs whenever the latest online reviews become available.

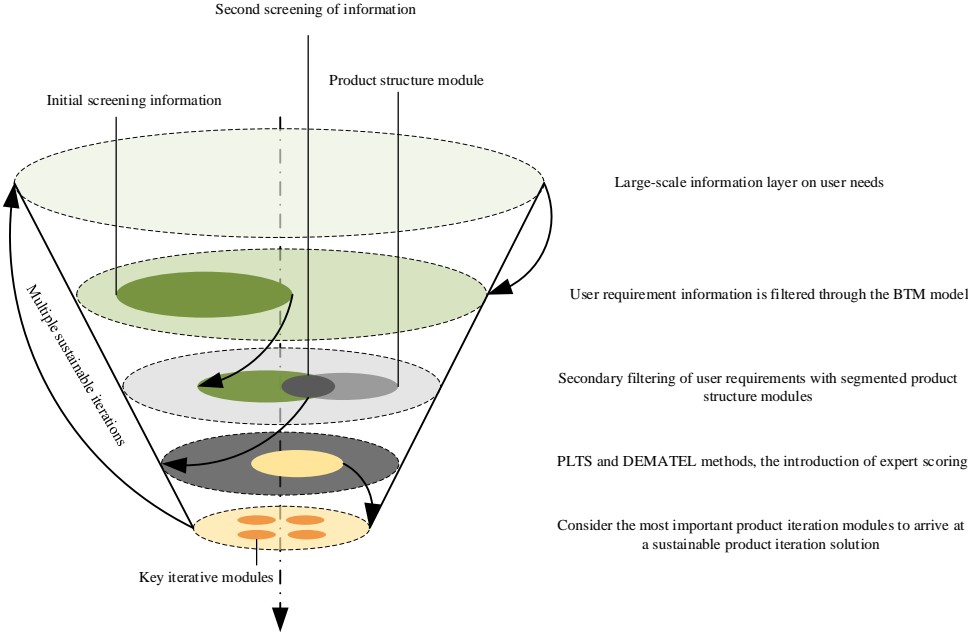

**Figure 1.** Flow chart of the sustainable product iterative design approach.

### 3.1. Mining User Needs Based on the BTM Model

Yan et al. [29] proposed a biterm topic model (BTM) to comprehensively explore the topics in a corpus and tackle the issue of sparsity in short texts. BTM posits that two words in a biterm share a common topic, which is derived from a hybrid topic across the corpus. In this framework, a topic is represented as a distribution of words. Unlike most topic models that examine the thematic factors of a corpus by modeling document generation, BTM achieves this objective by modeling the co-occurrence of word pairs. Specifically, if two phrases appear together more frequently, they are more likely to be related to the same topic [30]. As shown in Figure 2.

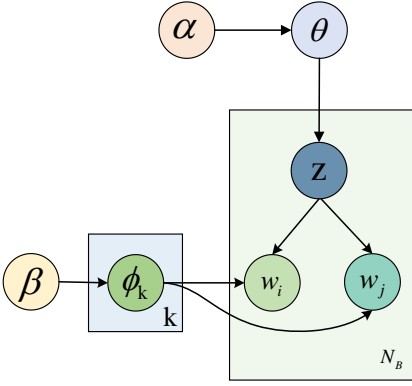

**Figure 2.** Schematic diagram of a BTM operation.

Assuming that a biterm is composed of words $w_{i,1}$ and $w_{i,2}$, that is, $b = \{w_{i,1}, w_{i,2}\}$, the two words in each biterm are sampled from the same topic $Z$. $N_B$ Biterms form a set

$B = \{b_i\}_{i=1}^{N_B}$. In addition, the symmetrical $\theta$ and $\phi_k$ in the Dirichlet prior are used, which have single-value hyperparameters $\alpha$ and $\beta$, respectively. The BTM is generated as follows:

1.  The topic distribution is generated for $\alpha$ with parameters in the Dirichlet prior, $\theta \sim Dir(\alpha)$;
2.  For each topic $K \in [1, k]$, the topic distribution is generated for $\beta$ with parameters in the Dirichlet prior, $\phi_k \sim \text{Dir}(\beta)$;
3.  For each biterm $b_i \in B$:

The topic distribution of the biterm $b_i$ in the topic distribution is $\theta$:$z_i \sim Multinomial(\theta)$. A word is generated according to the topic distribution $z_i$:$w_{i,1}, w_{i,2} \sim Multinomial(\phi_{z_i})$.

According to BTM, the probability of $b_i$ can be calculated by integrating $\theta$ and $\phi_k$. In addition, by multiplying the probability of each biterm, the entire corpus can be obtained. This can be stated as follows [30]:

$$P(B|\alpha, \beta) = \prod_{i=1}^{N_B} \iint \sum_{k=1}^{K} \theta_k \phi_{k,w_{i,1}} \phi_{k,w_{i,2}} \mathrm{d}\theta \mathrm{d}\Phi \tag{1}$$

*3.2. Evaluating User Requirements and Technical Modules using Probabilistic Semantic Term Sets*

Pang et al. [20] proposed the use of probabilistic linguistic term sets (PLTS) in their work. A PLTS is a semantic collection consisting of a set of semantic terms and their associated probabilistic information. In the case where the expert's evaluation of the semantic set represented by PLTS is denoted as *L(p)*, the corresponding definitions can be found.

In Definition 1 [31], suppose $S = \{s_{0,\dots,s_\alpha}\}$ is a semantic term set, $L(p)$ is called a PLTS on S:

$$L(p) = \left\{ L^{(k)} p^{(k)} \Big| L^{(k)} \in S, p^{(k)} \geq 0, k = 1, 2, \dots, \#L(p), \sum_{k=1}^{\#L(p)} p^{(k)} \leq 1 \right\} \tag{2}$$

In Equation (2), $\#L(p)$ denotes the number of semantic sets in the PLTS, and $L^{(k)}\left(p^{(k)}\right)$ represents the likelihood of the semantic set $L^{(k)} p^{(k)}$.

When $\sum_{k=1}^{\#L(p)} p^{(k)} = 1$, the semantic information in the experts' evaluation is complete; when $\sum_{k=1}^{\#L(p)} p^{(k)} < 1$, the semantic information in the experts' evaluation is an incomplete probability distribution and the PLTS needs to be standardized.

For example, assume that LTS is a semantic terminology set with five levels of granularity, $S^5 = (S_0 = VL, S_1 = L, S_2 = M, S_3 = H, S_4 = VH)$. When the evaluation indicator is the evaluation of the product's battery life, the user gives "The battery life of the product is good" as S4 with a probability of 0.6, and "I don't know much about the battery life" as S3 with a probability of 0.3. At this time, $L(p) = \{(S_1, 0.3), (S_4, 0.6)\}$. When the evaluation indicator evaluates the quality of the product, the first user gives a very good rating, the second user gives a very good rating, the third user gives a good rating, and the fourth an average rating, then the "quality of the product is very good" is $S_4$ and the probability is 0.5. The probability of a product being of good quality is $S_3$ and the probability is 0.25. The probability of the product being of average quality is $S_2$ and the probability is 0.25, and $L(p) = \{(S_2, 0.25), (S_3, 0.25), (S_4, 0.5)\}$.

In Definition 2 [31], suppose $L(p)$ is $S$ the incomplete PLTS in the previous semantic information, it is standardized to:

$$\overline{L}(p) = \left\{ L^{(k)} \overline{p}^{(k)} \Big| L^{(k)} \in S, k = 1, 2, \dots, \#L(p), \overline{p}^{(k)} = p^{(k)} \Big/ \sum_{k=1}^{\#L(p)} p^{(k)} \right\} \tag{3}$$

In Definition 3 [31], $L_1{}^{(k)}$ and $L_2{}^{(k)}$ are the K-th semantic sets in $L(p)_1$ and $L(p)_2$, $p_1{}^{(k)}$ and $p_2{}^{(k)}$ are the probability information of semantic sets $L_1{}^{(k)}$ and $L_2{}^{(k)}$, respectively, and the distance between $L(p)_1$ and $L(p)_2$ is:

$$d(L(p)_1, L(p)_2) = \sqrt{\sum_{k=1}^{\#L(p)_1} \left[ p_1{}^{(k)} r_1{}^{(k)} - p_2{}^{(k)} r_2{}^{(k)} \right]^2 / \#L(p)_1} \tag{4}$$

where $r_1{}^{(k)}$ and $r_2{}^{(k)}$ denote the subscripts of $L_1{}^{(k)}$ and $L_2{}^{(k)}$.

In Definition 4 [31], suppose $L(p)$ is the $S$ of the previous PLTS and $r^{(k)}$ is the subscript of the semantic term set $L^{(k)}$, then the score function, degree of deviation, and degree of hesitation of $L(p)$ are:

$$E(L(p)) = S_{\overline{\alpha}} \tag{5}$$

$$\sigma(L(p)) = \left[ \sum_{k=1}^{\#L(p)} \left( p^{(k)} (r^{(k)} - \overline{\alpha}) \right)^2 \right]^{1/2} \Bigg/ \sum_{k=1}^{\#L(p)} p^{(k)} \tag{6}$$

$$H(L(p)) = \frac{\frac{1}{\#L(p)} \sum_{k=1}^{\#L(p)} \left[ p^{(k)} (r^{(k)} - \overline{\alpha}) \right]^2}{\alpha + 1} \tag{7}$$

In Equation (8), $\overline{\alpha} = \sum_{k=1}^{\#L(p)} r^{(k)} p^{(k)} \Big/ \sum_{k=1}^{\#L(p)} p^{(k)}$. If the two PLTSs are $L(p)_1$ and $L(p)_2$, respectively, and if $SF(L(p_1)) > SF(L(p_2))$, then $L(p)_1$ is superior to $L(p)_2$.

In Definition 5 [31], suppose $L(p)$ is the S of the previous PLTS, then $L(p)$ is converted to an exact numerical function:

$$SF(L(p)) = \overline{\alpha} - \sigma(L(p)) - H(L(p)) \tag{8}$$

The greater the $\overline{\alpha}$, the smaller the degree of deviation; the smaller the degree of hesitation, the better is the PLTS $L(p)$.

*3.3. Calculating User Requirement Weights Using the Improved Probabilistic Semantic DEMATEL Method*

The next step uses the enhanced probabilistic semantic DEMATEL method for calculating the weights of indicators in multi-attribute decision making. In multi-attribute decision making, indicator interactions are typically considered when applying factors to the exact numerical DEMATEL method and to probabilistic semantic improvements. This involves converting the semantic variables into probabilistic semantics and analyzing the indicators' interactions using the DEMATEL method to determine their importance. Importance calculations are then performed to obtain the weights for each indicator. To calculate the index weight using the improved probabilistic semantic DEMATEL method, the following steps should be followed:

1.  A direct correlation matrix $X^k$ between the indexes was established. The LTS term collection for the correlation between the evaluation index is $S^r = \{s_g | g = 0, 1, 2, \ldots, e\}$. The evaluation indexes are $C_j (j = 1, 2, \ldots, n)$. Expert $E_k (1 \le k \le t)$ evaluation of the correlation between indexes, according to the collection Sr, was used to establish a direct correlation matrix between indexes.

$$X^k = \begin{bmatrix} 0 & s_{g(12)}^k & \cdots & s_{g(1n)}^k \\ s_{g(21)}^k & 0 & \cdots & s_{g(2n)}^k \\ \vdots & \vdots & & \vdots \\ s_{g(n1)}^k & s_{g(n2)}^k & \cdots & 0 \end{bmatrix} \tag{9}$$

where $s^k_{g(ij)}$ represents the degree of influence of the use of the LTS evaluation index $C_i$ by expert Ek on $C_j$. It takes a value of 0 if there is no influence.

2. All expert evaluations of the inter-influence of relationships between the indicators were assembled according to the example in Definition 1 to obtain a direct correlation matrix between the indicators in the form of a probabilistic semantic term set for all experts.

$$X = \begin{bmatrix} 0 & L(p)_{12} & \cdots & L(p)_{1n} \\ L(p)_{21} & 0 & \cdots & L(p)_{2n} \\ \vdots & \vdots & & \vdots \\ L(p)_{n1} & L(p)_{n2} & \cdots & 0 \end{bmatrix} \tag{10}$$

where $L(p)_{ij}$ represents the PLTS, and the exact numerical calculation is usually adopted during the calculation by the traditional DEMATEL method. The current data are still semantic information, which makes it impossible to calculate, so this study uses binary semantics to convert the semantic information into exact values. This is performed using Equations (5) and (6) to calculate the score function and the deviation degree $\sigma(L(p))$ of the probabilistic semantic $L(p)_{ij}$. Subsequently, binary semantics were used to convert them into exact numerical values.

In Definition 6, $\beta \in [0, g]$ is the result of the semantic term set L after integration, and let $i = round(\beta)$, $i \in [0, g]$. $\beta$ can be expressed by the following function $\Delta$ as a binary semantic conformance [31]:

$$\Delta[0, g] \rightarrow L \times [0.5, -0.5] \quad \Delta(\beta) = (l_i, \alpha_i)$$
$$\begin{cases} l_i & i = round(\beta) \\ \alpha_i = \beta - i & \alpha_i \in [-0.5, 0.5) \end{cases}$$

where *round* denotes the rounding operator, $\alpha$ denotes the transfer value. In contrast, let $(l_i, \alpha_i)$ be binary semantic information and the information is converted into a real number using a $\Delta^{-1}$ function.

$$\Delta^{-1} : L \times [0.5, -0.5] \rightarrow [0, g], \Delta^{-1}(l_i, \alpha_i) = i + \alpha_i = \beta$$

3. The score function and the degree of deviation were calculated and then converted into a direct correlation matrix after obtaining exact values using the $\Delta^{-1}$ function.

$$X = \begin{bmatrix} 0 & \Delta^{-1}(E(L(p)_{12}), \sigma(L(p)_{12})) & \cdots & \Delta^{-1}(E(L(p)_{1n}), \sigma(L(p))_{1n}) \\ \Delta^{-1}(E(L(p)_{21}), \sigma(L(p)_{21})) & 0 & \cdots & \Delta^{-1}(E(L(p)_{2n}), \sigma(L(p))_{2n}) \\ \vdots & \vdots & & \vdots \\ \Delta^{-1}(E(L(p)_{n1}), \sigma(L(p)_{n1})) & \Delta^{-1}(E(L(p)_{n2}), \sigma(L(p)_{n2})) & \cdots & 0 \end{bmatrix} \tag{11}$$

4. The directly normalized correlation matrices were then calculated. A common method for normalizing directly correlated matrices is based on the sum of the vector factors of every row of the matrix [32]. Let the normalization coefficient of X be $\lambda$, the normalization coefficient is calculated with the score function and the degree of deviation in the PLTS, the calculation form of $\lambda$ is calculated as follows:

$$\lambda = 1 / \max_{1 \leq i \leq n} \left( \sum_{j=1}^{n} \Delta^{-1}(E(L(P)_{ij}), \sigma(L(P)_{ij})) \right) \tag{12}$$

The normalized direct correlation matrix Z is:

$$Z = \lambda X \tag{13}$$

5. The total correlation matrix was calculated using *T*. According to references [33,34], *T* is calculated as follows:

$$T = \begin{bmatrix} 0 & t_{12} & \cdots & t_{1n} \\ t_{21} & 0 & \cdots & t_{2n} \\ \vdots & \vdots & & \vdots \\ t_{n1} & t_{n2} & \cdots & 0 \end{bmatrix}$$

$$T = Z(1-Z)^{-1} \tag{14}$$

6    Index importance $\omega_j$ was calculated.

The sum of $i$ in row $I$ of matrix $T$ is defined as $D_j$ and the sum of $j$ in column is defined as $F_j$.

$$D_j = (D_j)_{n \times 1} = \left[ \sum_{j=1}^{n} t_{ij} \right]_{n \times 1} \tag{15}$$

$$F_j = (F_j)_{n \times 1} = (F_j)'_{1 \times n} = \left[ \sum_{i=1}^{n} t_{ij} \right]'_{1 \times n} \tag{16}$$

Let $i = j$, the formula for calculating the index importance $\omega_j$ is as follows:

$$\omega_j = \sqrt{(D_j + F_j)^2 + (D_j - F_j)^2} \tag{17}$$

The index importance was normalized to obtain the index weight $\overline{\omega}_j (1 \le j \le n)$. The index weight set was $\overline{\omega}_j = W = [\overline{\omega}_1, \cdots, \overline{\omega}_j, \cdots, \overline{\omega}_n]$.

### 3.4. Sequencing Technology Modules with Consideration of User Requirements Interaction

Suppose that the number of experts is $E_k (1 \le k \le t)$, the expert weight is $O_k (1 \le k \le t)$, the user need is $R_j (1 \le j \le m)$, the technical module is $C_i (1 \le i \le n)$, and the weight of need is $W_j (1 \le j \le m)$.

The correlation between user needs $R_j$ and technical module $C_i$ is evaluated by the expert as:

$$G_{ij} = \sum_{k=1}^{t} \left[ SF(L(P)^k_{ij}) \cdot O_k \right] \tag{18}$$

The weight between user need $R_j$ is evaluated by the expert as:

$$H_j = \sum_{k=1}^{t} \left[ W_j \cdot O_k \right] \tag{19}$$

The importance of technical modules considering the mutual influence between user needs is:

$$C_i = \sum_{j=1}^{m} H_j \cdot G_{ij} \tag{20}$$

## 4. Case Studies

### 4.1. Cases

In "The Use of UAVs as Research Products: A Comparative Analysis of User Reviews on Jindong Shopping Platform and Their Application in Experimental Data Analysis," it was found that the versatility of UAVs (Unmanned Aerial Vehicles) was very attractive to customers and resulted in high sales volume [34]. In this study, we employed an UAV as the research product and conducted a comprehensive comparison of user reviews on the Jindong Shopping Platform (www.jd.com, accessed on 22 July 2022). The reviews were found to be relatively objective in content, abundant in information, and highly correlated with the products reviewed. A total of 15,303 reviews of a specific UAV product were collected and used as the data source for our experiment. We utilized Python web crawler technology to collect online reviews, including the review text, product type, review title,

and review time. After eliminating invalid information, such as repeated reviews, we conducted data pre-processing, which involved segmenting the valid reviews into words and sentences using the Python Jieba word segmentation package. The stop word list by the Harbin Institute of Technology (HIT) was used to filter out stop words, resulting in 9085 pieces of valid comment information. Finally, we sorted and generated a corpus of online reviews for further analysis.

First, the parameters were set. The setting of relevant parameters, including prior hyperparameters α, β, and the number of topics *k*, was completed before running the UAV reviews in the BTM model. Second, the $N_{iter}$ was set to 1000 times by default, according to the sample data. During model training, the results were saved after every 100 iterations. Repeated tests on the reviews of the UAV product showed that when the number of topics, *k*, was set to 12, the extraction effects were the best; α was set to 10, and β was set to 0.5. Finally, the content of each topic was inferred according to the topic clustering results and ranked by probability from high to low. Each topic's five high-frequency words and five low-frequency words were screened as keywords. The topic contents focused on product price, function, promotion, users' needs for the product itself, experience, and services, etc. Figure 3 presents the results for these topics.

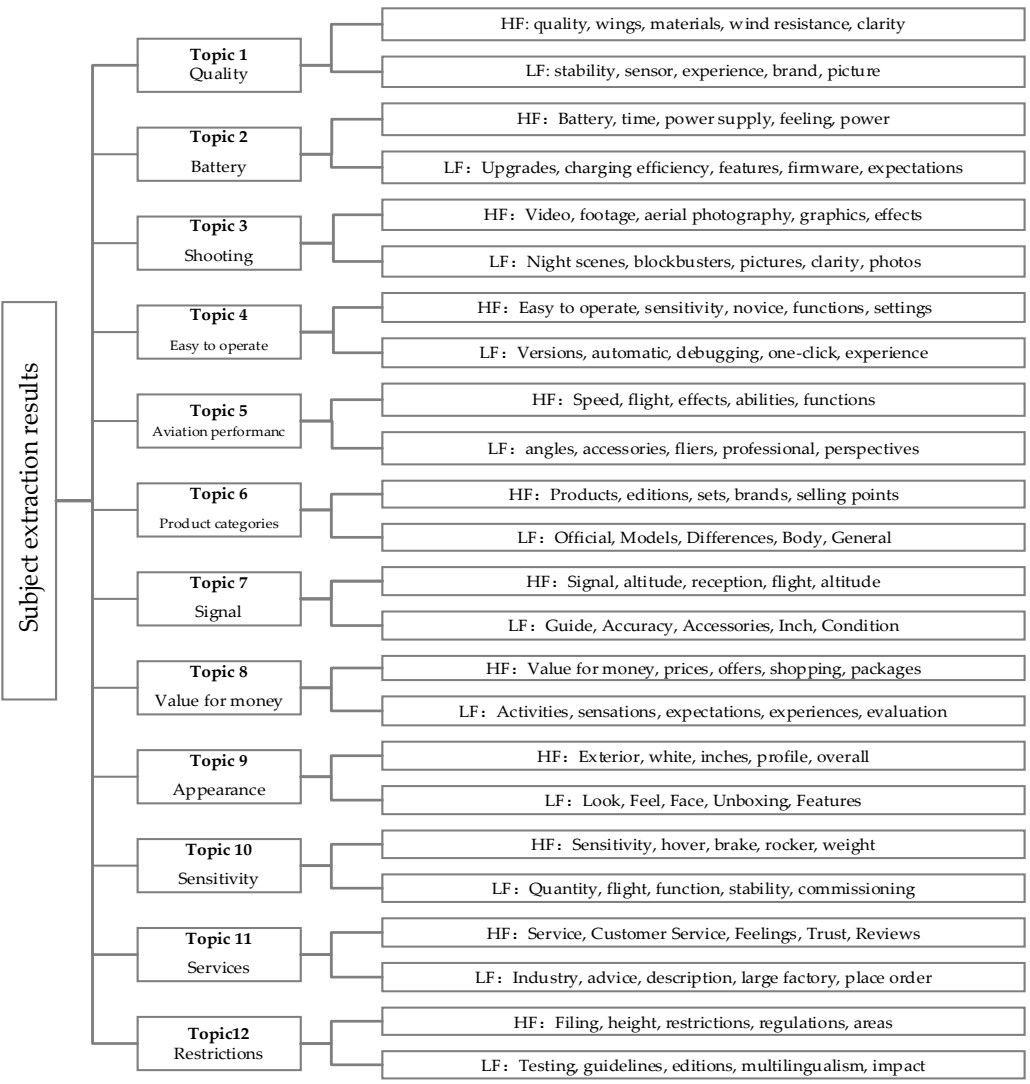

**Figure 3.** Subject extraction results.

1.  The user topics that needed to be extracted were divided by probability. The repeated user needs information was integrated and divided into six topics, from high to low:

quality ($R_1$), battery ($R_2$), shooting ($R_3$), convenient operation ($R_4$), signal ($R_5$), and cost performance ($R_6$), as shown in Figure 4.

**Figure 4.** User requirements filtering diagram.

2. The technical structure modules were divided by analyzing the patents of this UAV brand and the experts' advice. There were eight categories of technical modules: power module ($C_1$), including the motor; photography module ($C_2$), including photography component and picture transmission signal transmitter; gimbal module ($C_3$), including gimbal motor; control module ($C_4$), including remote control technology; interaction module ($C_5$), including monitor and control panel; flight module ($C_6$), including controller, propeller, and wing; efficiency module ($C_7$), including battery power cable; and carrier module ($C_8$), including the overall weight and volume of the UAV.

3. The evaluation and scoring were performed by an expert group. The expert group comprised six experts, including professional product structure designers, brand experts, and product development technicians. In this case, a collection of five-granularity semantic terms was defined: $S^5 = \{S_0 = $ Extremely low, $S_1 = $ low, $S_2 = $ medium, $S_3 = $ high, $S_4 = $ extremely high$\}$. Owing to limited space, the evaluation of only one expert, $E_1$, is presented herein. The evaluation indexes for user needs and technical structural modules by expert $E_1$ were converted into the form of a PLTS, as shown in Table 1. Expert $E_1$ converted the probabilistic semantic term evaluation of the user needs and technical modules into a score function, as shown in Table 2.

**Table 1.** Expert $E_1$ evaluation of the indicators relating user needs to the technology modules.

| | $C_1$ | $C_2$ | $C_3$ | $C_4$ | $C_5$ | $C_6$ | $C_7$ | $C_8$ |
|---|---|---|---|---|---|---|---|---|
| $R_1$ | {$S_3$(0.4), $S_4$(0.6)} | {$S_3$(0.3), $S_4$(0.7)} | {$S_3$(0.5), $S_4$(0.5)} | {$S_3$(0.4), $S_4$(0.6)} | {$S_2$(0.25), $S_3$(0.25), $S_4$(0.5)} | {$S_2$(0.3), $S_3$(0.7)} | {$S_3$(0.4), $S_4$(0.6)} | {$S_3$(0.4), $S_4$(0.6)} |
| $R_2$ | {$S_1$(1)} | {$S_1$(0.3), $S_2$(0.7)} | {$S_1$(0.5), $S_2$(0.5)} | {$S_4$(1)} | {$S_3$(0.4), $S_4$(0.6)} | {$S_2$(0.4), $S_3$(0.6)} | {$S_3$(0.5), $S_4$(0.5)} | {$S_3$(0.3), $S_4$(0.7)} |
| $R_3$ | {$S_2$(0.4), $S_3$(0.6)} | {$S_2$(0.25), $S_3$(0.5), $S_4$(0.25)} | {$S_2$(0.3), $S_3$(0.7)} | {$S_2$(0.2), $S_3$(0.8)} | {$S_2$(0.3), $S_3$(0.7)} | {$S_4$(1)} | {$S_2$(0.4), $S_3$(0.6)} | {$S_2$(0.5), $S_2$(0.5)} |
| $R_4$ | {$S_1$(0.4), $S_2$(0.6)} | {$S_1$(0.5), $S_2$(0.5)} | {$S_1$(0.4), $S_2$(0.6)} | {$S_0$(0.4), $S_1$(0.6)} | {$S_1$(0.2), $S_2$(0.8)} | {$S_3$(0.5), $S_4$(0.5)} | {$S_1$(0.2), $S_2$(0.8)} | {$S_0$(0.25), $S_1$(0.25), $S_2$(0.5)} |
| $R_5$ | {$S_3$(0.3), $S_4$(0.7)} | {$S_3$(1)} | {$S_0$(0.5), $S_1$(0.5)} | {$S_2$(0.4), $S_3$(0.6)} | {$S_1$(0.2), $S_2$(0.8)} | {$S_1$(0.3), $S_2$(0.7)} | {$S_2$(0.3), $S_3$(0.7)} | {$S_0$(0.2), $S_1$(0.8)} |
| $R_6$ | {$S_3$(1)} | {$S_3$(0.3), $S_4$(0.7)} | {$S_2$(0.25), $S_3$(0.25), $S_4$(0.5)} | {$S_3$(0.5), $S_4$(0.5)} | {$S_3$(0.4), $S_4$(0.6)} | {$S_3$(0.3), $S_4$(0.7)} | {$S_3$(0.4), $S_4$(0.6)} | {$S_3$(0.5), $S_4$(0.5)} |

**Table 2.** Expert $E_1$ score function for correlating user requirements with technical modules.

|  | $C_1$ | $C_2$ | $C_3$ | $C_4$ | $C_5$ | $C_6$ | $C_7$ | $C_8$ |
|---|---|---|---|---|---|---|---|---|
| $R_1$ | $S_{3.6}$ | $S_{3.7}$ | $S_{3.5}$ | $S_{3.6}$ | $S_{3.25}$ | $S_{2.7}$ | $S_{3.6}$ | $S_{3.6}$ |
| $R_2$ | $S_1$ | $S_{1.7}$ | $S_{1.5}$ | $S_4$ | $S_{3.6}$ | $S_{2.6}$ | $S_{3.5}$ | $S_{3.7}$ |
| $R_3$ | $S_{2.6}$ | $S_3$ | $S_{2.7}$ | $S_{2.8}$ | $S_{2.7}$ | $S_4$ | $S_{2.6}$ | $S_2$ |
| $R_4$ | $S_{1.6}$ | $S_{1.5}$ | $S_{1.6}$ | $S_{0.6}$ | $S_{1.8}$ | $S_{3.5}$ | $S_{1.8}$ | $S_{1.25}$ |
| $R_5$ | $S_{3.7}$ | $S_3$ | $S_{0.5}$ | $S_{2.6}$ | $S_{1.8}$ | $S_{1.7}$ | $S_{2.7}$ | $S_{0.8}$ |
| $R_6$ | $S_3$ | $S_{3.7}$ | $S_{3.25}$ | $S_{3.5}$ | $S_{3.6}$ | $S_{3.7}$ | $S_{3.6}$ | $S_{3.5}$ |

The degree of deviation between the expert $E_1$ evaluation and the correlation between user needs and technical modules can be calculated using Equation (6), as shown in Table 3.

**Table 3.** Degree of deviation of expert $E_1$ from the correlation between user requirements and technical modules.

|  | $C_1$ | $C_2$ | $C_3$ | $C_4$ | $C_5$ | $C_6$ | $C_7$ | $C_8$ |
|---|---|---|---|---|---|---|---|---|
| $R_1$ | 0.3394 | 0.297 | 0.3535 | 0.3394 | 0.1822 | 0.2969 | 0.3394 | 0.3394 |
| $R_2$ | 0 | 0.297 | 0.3535 | 0 | 0.3394 | 0.3394 | 0.3535 | 0.2969 |
| $R_3$ | 0.3394 | 0.354 | 0.2969 | 0.2262 | 0.2969 | 0 | 0.3394 | 0.3535 |
| $R_4$ | 0.3394 | 0.354 | 0.3394 | 0.3394 | 0.2262 | 0.3535 | 0.2262 | 0.5901 |
| $R_5$ | 0.2969 | 0 | 0.3535 | 0.3394 | 0.2262 | 0.2969 | 0.2969 | 0.2262 |
| $R_6$ | 0 | 0.2969 | 0.1822 | 0.3535 | 0.3394 | 0.2969 | 0.3394 | 0.3535 |

The correlation between the expert and user needs and technical modules can be converted into exact numerical values using Equation (18). Subsequently, six experts ($E_1$–$E_6$) were averaged as shown in Table 4.

**Table 4.** Precise function values of user requirements and technical modules by six experts.

|  | $C_1$ | $C_2$ | $C_3$ | $C_4$ | $C_5$ | $C_6$ | $C_7$ | $C_8$ |
|---|---|---|---|---|---|---|---|---|
| $R_1$ | 3.3217 | 3.3217 | 3.1340 | 2.8217 | 3.0656 | 2.3943 | 3.2491 | 3.2491 |
| $R_2$ | 1.0670 | 1.3943 | 1.1340 | 3.3191 | 3.2491 | 1.6915 | 3.1340 | 3.3943 |
| $R_3$ | 2.2491 | 2.6382 | 2.3943 | 2.4089 | 2.3943 | 3.5670 | 2.2491 | 1.6340 |
| $R_4$ | 1.2491 | 1.1340 | 1.3217 | 0.2491 | 0.9089 | 3.1340 | 1.5687 | 0.6367 |
| $R_5$ | 2.5141 | 2.6971 | 0.1340 | 2.2491 | 1.0687 | 1.3943 | 2.3943 | 0.5687 |
| $R_6$ | 3.1971 | 3.3943 | 3.0656 | 3.1340 | 3.2491 | 3.3943 | 3.2491 | 3.1340 |

Because there are a few repeated keywords in the BTM model topics, the mutual influence between user needs was considered in this study to obtain further user need information. The evaluation of the mutual influence between user needs was considered by expert $E_1$ and was converted into the form of a PLTS, as shown in Table 5. The evaluation of the mutual influence between user needs by expert $E_1$ was converted into a score function, as shown in Table 6.

**Table 5.** Expert $E_1$ evaluation of the probabilistic semantics terminology for the impact between user requirements.

|  | $R_1$ | $R_2$ | $R_3$ | $R_4$ | $R_5$ | $R_6$ |
|---|---|---|---|---|---|---|
| $R_1$ | 0 | {$S_2(0.75)$, $S_3(0.25)$} | {$S_3(0.2)$, $S_4(0.8)$} | {$S_0(0.1)$, $S_1(0.5)$, $S_2(0.4)$} | {$S_3(0.4)$, $S_4(0.6)$} | {$S_3(0.75)$, $S_4(0.25)$} |
| $R_2$ | {$S_2(0.25)$, $S_3(0.5)$, $S_4(0.25)$} | 0 | {$S_3(0.5)$, $S_4(0.5)$} | {$S_3(0.75)$, $S_4(0.25)$} | {$S_0(0.1)$, $S_1(0.6)$, $S_2(0.3)$} | {$S_1(0.25)$, $S_2(0.5)$, $S_3(0.25)$} |
| $R_3$ | {$S_1(0.2)$, $S_2(0.8)$} | {$S_3(0.5)$, $S_4(0.5)$} | 0 | {$S_2(0.3)$, $S_3(0.7)$} | {$S_0(0.3)$, $S_1(0.7)$} | {$S_3(0.25)$, $S_4(0.75)$} |
| $R_4$ | {$S_3(0.5)$, $S_4(0.5)$} | {$S_2(0.25)$, $S_3(0.75)$} | {$S_2(0.2)$, $S_3(0.8)$} | 0 | {$S_0(0.2)$, $S_1(0.8)$} | {$S_1(0.25)$, $S_2(0.75)$} |
| $R_5$ | {$S_3(0.75)$, $S_4(0.25)$} | {$S_1(0.25)$, $S_2(0.75)$} | {$S_1(0.25)$, $S_2(0.5)$, $S_3(0.25)$} | {$S_1(0.25)$, $S_2(0.75)$} | 0 | {$S_0(0.5)$, $S_1(0.5)$} |
| $R_6$ | {$S_3(0.5)$, $S_4(0.5)$} | {$S_1(0.3)$, $S_2(0.7)$} | {$S_0(0.1)$, $S_1(0.9)$} | {$S_0(0.2)$, $S_1(0.8)$} | {$S_2(0.1)$, $S_3(0.9)$} | 0 |

**Table 6.** Expert $E_1$ score function for the interconnection between user requirements.

| | $R_1$ | $R_2$ | $R_3$ | $R_4$ | $R_5$ | $R_6$ |
|---|---|---|---|---|---|---|
| $R_1$ | 0 | $S_{2.25}$ | $S_{3.8}$ | $S_{1.4}$ | $S_{3.6}$ | $S_{3.25}$ |
| $R_2$ | $S_3$ | 0 | $S_{3.5}$ | $S_{3.25}$ | $S_{1.2}$ | $S_2$ |
| $R_3$ | $S_{1.8}$ | $S_{3.5}$ | 0 | $S_{2.7}$ | $S_{0.7}$ | $S_{3.75}$ |
| $R_4$ | $S_{3.5}$ | $S_{2.75}$ | $S_{2.8}$ | 0 | $S_{0.8}$ | $S_{1.75}$ |
| $R_5$ | $S_{3.25}$ | $S_{1.75}$ | $S_2$ | $S_{1.75}$ | 0 | $S_{0.5}$ |
| $R_6$ | $S_{3.5}$ | $S_{1.7}$ | $S_{0.9}$ | $S_{0.8}$ | $S_{2.9}$ | 0 |

The degree of deviation between expert $E_1$'s evaluation and user needs can be calculated using Formula (6), as shown in Table 7.

**Table 7.** The extent to which Expert $E_1$ deviates from the interconnectedness of user needs.

| | $R_1$ | $R_2$ | $R_3$ | $R_4$ | $R_5$ | $R_6$ |
|---|---|---|---|---|---|---|
| $R_1$ | 0 | 0.2651 | 0.2262 | 0.5288 | 0.3394 | 0.2651 |
| $R_2$ | 0.3535 | 0 | 0.3535 | 0.2651 | 0.2687 | 0.3125 |
| $R_3$ | 0.2262 | 0.3535 | 0 | 0.2969 | 0.2969 | 0.2651 |
| $R_4$ | 0.3535 | 0.2651 | 0.2262 | 0 | 0.2262 | 0.2651 |
| $R_5$ | 0.2651 | 0.2651 | 0.3125 | 0.2651 | 0 | 0.3535 |
| $R_6$ | 0.3535 | 0.2969 | 0.1272 | 0.2262 | 0.1272 | 0 |

The normalization coefficient $\lambda = 15.9246$ can be calculated using Formula (12). Subsequently, the total correlation matrix T can be calculated using Equations (13), (14), and (19), as shown in Table 8.

**Table 8.** Total correlation matrix $T$.

| | $R_1$ | $R_2$ | $R_3$ | $R_4$ | $R_5$ | $R_6$ |
|---|---|---|---|---|---|---|
| $R_1$ | 0.9957 | 0.9824 | 1.0991 | 0.8448 | 0.9006 | 1.0155 |
| $R_2$ | 1.1216 | 0.8130 | 1.0596 | 0.8897 | 0.7452 | 0.9305 |
| $R_3$ | 1.0181 | 0.9631 | 0.8097 | 0.8258 | 0.6901 | 0.9623 |
| $R_4$ | 1.0656 | 0.9035 | 0.9542 | 0.6484 | 0.6748 | 0.8518 |
| $R_5$ | 0.9151 | 0.7394 | 0.7993 | 0.6598 | 0.5155 | 0.6756 |
| $R_6$ | 0.9319 | 0.7260 | 0.7290 | 0.6009 | 0.6875 | 0.6078 |

The index weight based on the PLTS DEMATEL can be calculated according to Equations (15)–(17), as shown in Table 9.

**Table 9.** DEMATEL indicator weights based on the probabilistic semantic term.

| | $D_j$ | $F_j$ | $D_j + F_j$ | $D_j - F_j$ | $\omega_j$ | $\overline{\omega_j}$ |
|---|---|---|---|---|---|---|
| $R_1$ | 5.8380 | 6.0479 | 11.8859 | 0.2099 | 11.8878 | 0.1956 |
| $R_2$ | 5.5596 | 5.1273 | 10.6868 | −0.4323 | 10.6956 | 0.1760 |
| $R_3$ | 5.2691 | 5.4509 | 10.7200 | 0.1818 | 10.7216 | 0.1764 |
| $R_4$ | 5.0983 | 4.4694 | 9.5677 | −0.6290 | 9.5884 | 0.1578 |
| $R_5$ | 4.3046 | 4.2137 | 8.5184 | −0.0909 | 8.5189 | 0.1402 |
| $R_6$ | 4.2830 | 5.0435 | 9.3265 | 0.7605 | 9.3575 | 0.1540 |

According to Table 9, the normalized maximum reference weight is $\omega_r = \omega_1 = 0.195621$. The average normalized weights of all the experts are listed in Figure 5.

According to Figure 5, this can be obtained by normalizing the index importance as $\overline{\omega}_{1r} = 0.196792$, $\overline{\omega}_{2r} = 0.178659$, $\overline{\omega}_{3r} = 0.173552$, $\overline{\omega}_{4r} = 0.156606$, $\overline{\omega}_{5r} = 0.138382$, and $\overline{\omega}_{6r} = 0.156009$. Finally, the importance of the technology module, considering the influence between user requirements, is obtained by using Equation (20) combined with calculating the data in Table 4 and Figure 5, $C = (2.277, 2.441, 1.939, 2.406, 2.396, 2.606, 2.674, 2.197)^T$.

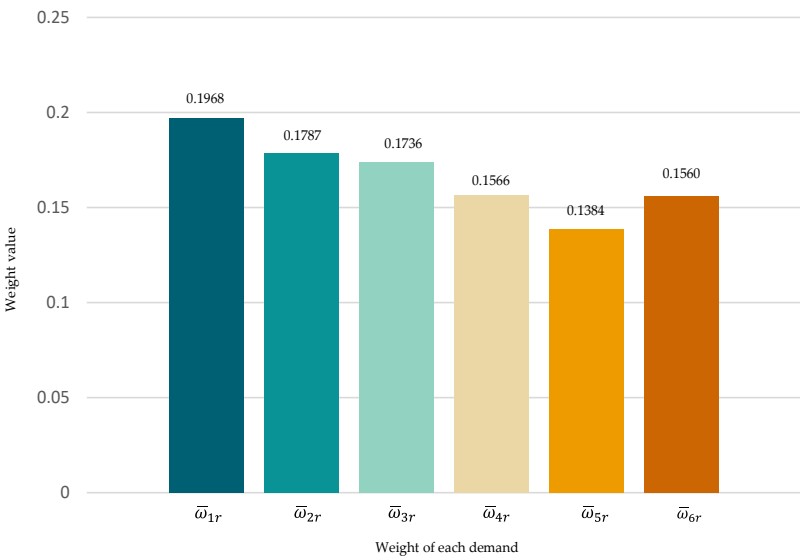

**Figure 5.** Normalized mean weights of the six experts.

*4.2. Experimental Results*

The importance of the technical modules that influenced user needs were calculated by combining the results of all experts with Formula (20) and resulted in: $C_1(2.277)$, $C_2(2.441)$, $C_3(1.939)$, $C_4(2.406)$, $C_5(2.396)$, $C_6(2.606)$, $C_7(2.674)$, and $C_8(2.197)$, respectively. The order of importance (from high to low) was $C_7 > C_6 > C_2 > C_4 > C_5 > C_1 > C_8 > C_3$. The efficiency ($C_7$), flight ($C_6$), photography ($C_2$), and control ($C_4$) were the four most important modules. Therefore, priority was given to the iterative improvement of these four modules in the UAV component.

By conducting a focus group to assess user concerns related to UAV components and focusing on technical structures and functions, the upgrade of UAVs can be significantly advanced. Additionally, this approach can ensure the sustainable production and utilization of UAV components, thereby promoting the sustainable development of UAVs. Figure 6 displays the UAV module components that were analyzed in combination with their functional characteristics, based on meeting user needs.

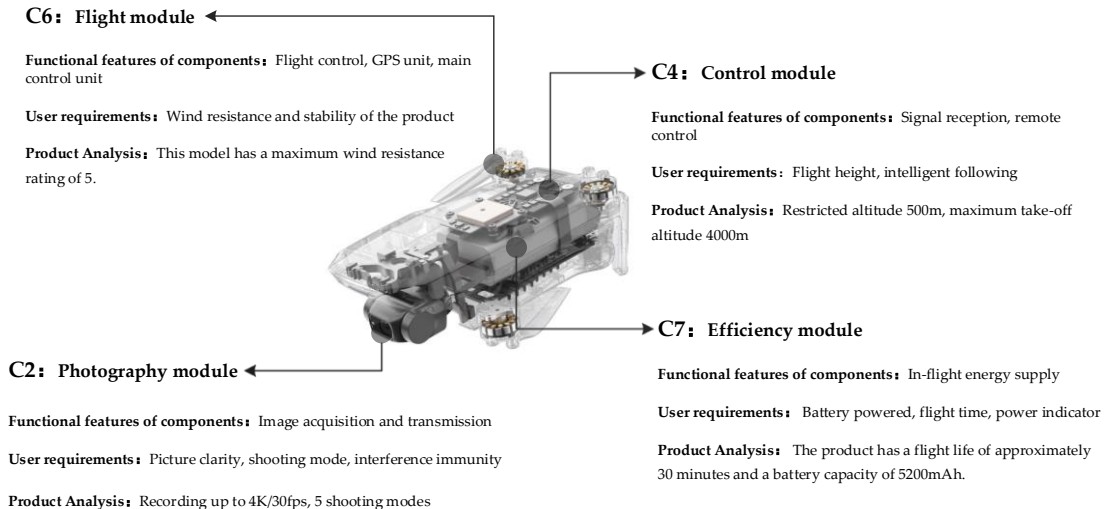

**Figure 6.** Analysis of the important component modules.

This approach places the structure at the core, with a focus on the main component modules and user needs. Based on an analysis of the critical component modules, the primary structural components that met user needs were determined to be the battery

module, body shell, propeller, shooting lens, fixed flight path data, and signal receiver. Following the identification of the iterative components, a focus group discussion was conducted to gather feedback on product improvement, as presented in Table 10.

**Table 10.** Product iterative improvement design scheme.

| Index | User Requirements | Product Components | Focus Group Improvement Programme | Sustainable Development Requirement |
|-------|-------------------|--------------------|-----------------------------------|-------------------------------------|
| 1 | Battery power, flight duration | Battery modules | Improving battery charging efficiency and battery capacity. | It is recommended to use batteries that do not contain harmful substances, such as lithium iron phosphate, lithium polymer battery, lithium polymer battery, etc |
| 2 | Wind resistance and stability of drones | Fuselage shells, propellers | Lifting of the overall UAV housing counterweight to ensure flight; widening of the propeller to adjust propeller orientation in the event of wind. | The overall materials of the drone body, such as polypropylene, polyamide and polycarbonate, should meet the specific environmental regulations of developing countries |
| 3 | Picture clarity | Shooting footage | Professional photographers, with the possibility of configuring higher photographic components, offering individual choice of parts. | Photographic components shall meet the technical requirements of environmental labeling products |
| 4 | photography module | Fixed flight path data | Professional photographers, with the possibility of configuring higher photographic components, offering individual choice of parts. | This demand is not sustainable development requirements |
| 5 | Smart Follow | Signal Receiver, Recording Lens | Set receiver module, drone sensor receiver module, intelligent following, more suitable for the elderly and children to go out to monitor and professional record life bloggers. | This demand is not sustainable development requirements |

## 5. Conclusions and Outlook

In this paper, we proposed a sustainable iterative research method for mining user needs and integrating decision-making products based on online reviews and utilizing probabilistic semantic term sets (PLTS). Our method leverages expert evaluations to address the hesitation and uncertainty that exists between user requirements and product modules, and uses the DEMATEL method to consider the correlation between requirement indicators. By improving the association function of DEMATEL using binary semantics, we reconstructed the PLTS acquisition score function with a deviation into an exact value DEMATEL association function, thereby providing accurate feedback on how users feel about the use of product components while ensuring the sustainable recycling of most product components.

Our research has three main innovations:

1.  The introduction of multi-attribute decision making into sustainable product iterative design, which ensures longer product life cycles, reduces the costs and risks for small and medium-sized manufacturing industries, and minimizes the waste of available resources.
2.  The improvement of the association function of DEMATEL, which accurately expresses the hesitation and uncertainty of expert evaluations, and solves the problem traditional multi-attribute decision making makes by not considering the interaction between indicators.

3. The improvement of the technical method of mining user demand information by combining it with the product structure module, which enables more accurate user demand screening and the identification of product components and modules for improvement using online reviews' information about user requirements.

Our experiments have shown that our method of user demand mining allows companies to accurately identify iterative product component modules that can be updated to meet user demand while ensuring the sustainable recycling of most components, and can compete with similar products by predicting future product development through constantly updated online reviews.

This paper has certain limitations, as the case product analyzed was a single type and may have a limited area of application, affecting the universality of the method. Future research will cover a variety of product types to expand the scope of this method. At the same time, in the product iteration process there are many manual steps which affect the result. We will build on this research to develop applications to improve the automation of sustainable iterative product design processes.

In future work, we will explore the generalization of our method to different product types and investigate its applicability to diverse industries. We will also investigate the use of other techniques for mining user needs, and explore the impact of incorporating additional sources of data into our method.

**Author Contributions:** Research methodology, Q.W., S.W. and S.F.; software, S.W.; validation, S.F. and S.W.; formal analysis, Q.W.; survey, S.W.; data collation, S.W.; writing—original draft preparation, S.F.; writing—review and editing, S.W.; supervision, Q.W.; project administration, S.W.; funding acquisition, Q.W. and S.F. All authors have read and agreed to the published version of the manuscript.

**Funding:** This study was supported by the Ministry of Education of China, Humanities and Social Science Foundation Research Fund (No. 21YJC760081).

**Institutional Review Board Statement:** Not applicable.

**Informed Consent Statement:** Not applicable.

**Data Availability Statement:** All data included in this study are available upon request by contacting the corresponding author.

**Conflicts of Interest:** The authors declare there are no conflict of interest.

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
