# Peer review of "A Sustainable Iterative Product Design Method Based on Considering User Needs from Online Reviews"

_sustainability, doi:10.3390/su15075950_

Round 1

Reviewer 1 Report

1. Compare to other MCDM techniques, why authors selected DEMATEl methodology in this research work, Need justification 

2. Authors can explain, How this study findings can be generalized and applicable to under developing countries for implementation 

3. Authors can mention, how to develop the product like software/app based on this research work for automating the sustainable Iterative Product Design process for the e-commerce companies 

Author Response

Dear Reviewer,

Thank you for your letter and for the reviewers’ comments concerning our manuscript entitled “A Sustainable Iterative Product Design Method Based on Considering User Needs from Online Reviews”. Those comments are all valuable and very helpful for revising and improving our paper, as well as the important guiding significance to our research. We have studied the comments carefully and have made a correction which we hope meets with approval. Revised portions are marked in red on the paper. The major corrections in the paper and the responses to the reviewer's comments are as flowing:

Responds to the reviewer's comments:

Point 1: Compare to other MCDM techniques, why authors selected DEMATEl methodology in this research work, Need justification

Response 1: We would like to thank you for your professional comments on our articles. In the process of multi-attribute decision-making, the traditional index weight calculation method usually determines the weight reflecting the characteristics of the indicators according to subjective or objective methods but ignores the correlation between indicators. In fact, the indicators are complementary, cross, conflict, and so on. For example, in the evaluation and selection of UAV products, users usually consider product quality, service life, functions, etc., while the life cycle is usually shorter if the product cost index is low.Decision making trial and evaluation laboratory (DEMATEL) method is often used to analyze the correlation between indicators.

For the evaluation index correlation and business evaluation information in the form of semantic phrases given by Wang et al., the DEMATEL method is used to determine the index weight and the improved TOPSIS method is used to select the best business models.

“Wang Fayu, Jiang Yan. Earning interest analysis of users in campus wireless network based on self-organizing neural network and fuzzy C-means clustering algorithmï¼»J].Application Research of Computers,2018,35( 1): 186-189. (In Chinese)”

Luo et al. adopted the interval 2 fuzzy DEMATEL method to analyze the interdependence among indicators, which is used to reduce the inherent complexity of decision-making problems, and obtained the ranking of alternative schemes by the hierarchical interval 2 fuzzy TOPSIS method.

“Luo Ling, Yang You,Ma Yan. Research on online collaborative learning group division based on fuzzy C-means algorithmï¼»J]. Computer Engineering and Applications,2017,53( 16): 68-73. (In Chinese)”

In this paper, the probabilistic semantic DEMATEL method is used to calculate the normalization and total matrix of the correlation matrix between indicators. Due to the complexity of the calculation, the score function of the correlation relationship between indicators is calculated first, and then the score function is converted to the exact value through the improved binary semantics. The DEMATEL method analyzes the correlation between indicators, obtains the weight of indicators, and reflects the correlation between indicators in the multi-attribute decision-making process.

Point 2: Authors can explain, How this study findings can be generalized and applicable to under developing countries for implementation

Response 2: We would like to thank the reviewers for their professional comments. At the beginning of this study, the problem of product iteration was accidentally discovered in cooperation with the small and medium manufacturing industries, so this study came into being. In the future, we are committed to first serving the cooperative small and medium-sized manufacturing industries, and constantly improving and perfecting the research results. Finally, the small and medium-sized manufacturing industries will promote the research results to developing countries.

Point 3: Authors can mention, how to develop the product like software/app based on this research work for automating the sustainable Iterative Product Design process for the e-commerce companies

Response 3: Once again, we would like to express our sincere thanks to the reviewers for their professional comments. We think this is an excellent suggestion. Paragraph 469-471: In this part, we will add explanations to the problems ignored in this paper. Just like the reviewer's opinion, the current stage of our research is to develop application software based on this result.Specific additions are as follows:

“At the same time, in the process of product iteration, there are many manual steps, which will affect the result. We will build on this research in the future to develop applications to improve the automation of sustainable iterative product design processes.”

Finally, we would like to express our sincere thanks to the reviewers for their valuable comments, which greatly improved the quality of our papers.

Thank you and best regards.

Yours sincerely,

Si Fu                                                                                                                     

Reviewer 2 Report

1- The Related Work section can be written more comprehensively and include more recent references.

2- Provide a valid reference for figure 2.

3- Provide valid references for all Equations taken from the literature.

4- In the study section, you said you eliminated invalid information. What are the characteristics of this information? Why were they removed? Do not affect the results?

5- Throughout the text, use a comma for the noun phrase “a sustainable, iterative approach”

Author Response

Dear Reviewer,

Thank you for your letter and for the reviewers’ comments concerning our manuscript entitled “A Sustainable Iterative Product Design Method Based on Considering User Needs from Online Reviews”. Those comments are all valuable and very helpful for revising and improving our paper, as well as the important guiding significance to our research. We have studied the comments carefully and have made a correction which we hope meets with approval. Revised portions are marked in red on the paper. The major corrections in the paper and the responses to the reviewer's comments are as flowing:

Responds to the reviewer's comments:

Point 1:The Related Work section can be written more comprehensively and include more recent references.

Response 1:We sincerely appreciate the valuable comments. As suggested by the reviewer, we add references 9, 11, and 12 to supplement the relevant working sections. In lines 70-80, we have made supplementary explanations for related work according to the reviewer's comments. Specific added references are as follows:

“9.   Liu, Y.; Gan, W. X.; Zhang, Q., Decision-making mechanism of online retailer based on additional online comments of consumers. Journal of Retailing and Consumer Services 2021, 59.

  1. Yang, C.; Wu, L. G.; Tan, K.; Yu, C. Y.; Zhou, Y. L.; Tao, Y.; Song, Y., Online User Review Analysis for Product Evaluation and Improvement. Journal of Theoretical and Applied Electronic Commerce Research 2021, 16, (5), 1598-1611.”
  2. Yu, Z. Y.; Zhao, W.; Guo, X.; Hu, H. C.; Fu, C.; Liu, Y., Multi-Indicators Decision for Product Design Solutions: A TOPSIS-MOGA Integrated Model. Processes 2022, 10, (2).”

Point 2: Provide a valid reference for figure 2.

Response 2: We would like to thank you for your professional comments on our articles. According to the reviewer's suggestion, we have added references in line 185 of the article. For your convenience, we have marked the revised content in the article.

Point 3: Provide valid references for all Equations taken from the literature.

Response 3: We would like to thank you again for your suggestions on our articles. According to the suggestions of reviewers, we have added references to the definition in the article. They appear on lines 197, 215, 217, 221, and 226, respectively. For your convenience, we have marked the revised content in the article.

Point 4: In the study section, you said you eliminated invalid information. What are the characteristics of this information? Why were they removed? Do not affect the results?

Response 4: We thank the reviewers for their careful reading of our articles. In the research part, we deleted invalid information, including repeated comments, garbled characters, and so on. This section of the review is invalid information or reviews written repeatedly by the business to get good reviews and does not provide us with any valuable information about the user's needs. These comments are not what our real users think. So, it doesn't affect our results.

Point 5:Throughout the text, use a comma for the noun phrase “a sustainable, iterative approach”

Response 5: Thanks for your careful checks. Based on your comments, we have made the corrections to make the word harmonized within the whole manuscript. We modify the words that appear in the article on lines 11 and 132. For your convenience, we have marked the revised content in the article.

Finally, we would like to express our sincere thanks to the reviewers for their valuable comments, which greatly improved the quality of our papers.

Thank you and best regards.

Yours sincerely,

Si Fu

Author Response

Dear Reviewer,

Thank you for your letter and for the reviewers’ comments concerning our manuscript entitled “A Sustainable Iterative Product Design Method Based on Considering User Needs from Online Reviews”. Those comments are all valuable and very helpful for revising and improving our paper, as well as the important guiding significance to our research. We have studied the comments carefully and have made a correction which we hope meets with approval. Revised portions are marked in red on the paper. The major corrections in the paper and the responses to the reviewer's comments are as flowing:

Responds to the reviewer's comments:

Point 1: Paragraphs 9-10: There is an error in the sentence below. The word »my« is out of context.

“Small and medium-sized manufacturing industries can use online reviews to my valuable  user requirements, enabling them to iteratively and precisely upgrade their products based on user needs.”

Response 1: We sincerely thank the reviewer for careful reading. As suggested by the reviewer, we have removed the "my" from line 9. Thanks for your correction. In order to facilitate your browsing, the specific modifications are as follows:

“Small and medium-sized manufacturing industries can use online reviews to valuable user requirements, enabling them to iteratively and precisely upgrade their products based on user needs.”

Point 2: Paragraph 79: In this sentence” In the face of rapid economic development, decision-makers increasingly use a variety of semantic terms to provide evaluation information”, it is not clear in what context semantic terms were used to provide valuation information. To evaluate product characteristics? Additional words are missing to explain this usage.

Response 2: We sincerely thank you again for your careful inspection. We are sorry for our carelessness. According to your comments, we have made a correction. We have added "on product characteristics" in line 88. In order to facilitate your browsing, the specific modifications are as follows:

“In the face of rapid economic development, decision-makers increasingly use a variety of semantic terms to provide evaluation information on product characteristics.”

Point 3: Paragraph 289: Section 4.1. adequately describes the process for collecting reviews and categorizing them, but I cannot find any information about what you did with reviews that are not directly related to product features. E.g., user comments on the packaging, etc. Provide this information. In addition, how many products were included in this study?

Response 3: We greatly appreciate your professional comments on our articles. As you are concerned, only one UAV product is involved in this study. Second, I'd like to explain to you that we also categorize reviews that are not directly related to product features. You can see that we have divided online reviews into 12 topics, for example, reviews of products that users think are beautifully packaged. The first time we filter information, we might filter it down to the topic of appearance or service. Which topic depends on how often the comment appears and what words appear next to it. Unlike most topic models, which learn topic components in the corpus by modeling generated documents, BTM models complete this task by modeling the generation of contra point words, that is, if two words appear together more frequently, they are more likely to belong to the same topic. Therefore, we can screen out the reviews that are relevant to the product's features.

Finally, as the 15,303 comments we obtained are in Chinese with a large amount of information, we hope you can understand. Of course, if you still need to provide this information, we are more than happy to provide it to you.

Point 4: In the section "4.2. Experimental Results" it is not described what the sustainability criteria were. What methods of sustainability assessment did you use to decide which component was more sustainable than others?

Response 4: Once again, we would like to express our sincere thanks to the reviewers for their professional comments. We think this is an excellent suggestion. We have modified Table 12 according to the reviewer's suggestion and added the sustainable development requirements for the product iterative design scheme. Among them, sustainable development requires that it be assessed against criteria set by developing countries.

After our expert team knows the exact improved product modules, we evaluate them on a component-by-component basis, mainly on material sustainability. For the convenience of your browsing, the modified table 12 is as follows:

Table 12. Product iterative improvement design scheme

Finally, we would like to express our sincere thanks to the reviewers for their valuable comments, which significantly improved the quality of our papers.

Thank you and best regards.

Yours sincerely,

Si Fu

Reviewer 4 Report

This paper hopes to evaluate the indicators between experts' hesitation and uncertainty through the proposed a Sustainable Iterative Product Design Method, and use decision-making experiments and evaluation laboratory methods to analyze the correlation between demand indicators, thereby increasing the life of the product Cycle and reduce the waste of a lot of resources in the design process, and take drone products as an example. The motivation and purpose of the research fit for the theme goal of the journal, but whether the final research results are credible or not will have a key impact due to the characteristics of the product. To evaluate the use requirements, the relationship between the product life cycle and the design decision-making process, and drones may not necessarily be the best choice. Because the life cycle of the drone itself is not short as a phone, and online reviews are not intuitive, and it is even more difficult to convey the user experience in detail. Wearable devices, or products used in daily life may be more suitable as research and analysis targets, unless the author can provide more research data and information to explain the suitability of the research object. The author should add whether the product life cycle of drones is short and cite information such as the huge sales volume of the product to prove the suitability of the product category selection. In addition, there are a lot of numbers and calculation formulas in this article. In addition to presenting the numbers themselves, these econometric studies should also be supplemented with some corresponding visual charts to display the research results,, such as bar charts and pie charts , or a regression distribution curve maps. This article focuses on design research, the way and level of image presentation should also be paid attention to, and the quality of image expression should also be improved. 

Although the research motivation of this article is correct and good, but in terms of research rigor, data reliability and graphic quality, the reviewers suggest that the authors should improve the graphics to fully meet the basic requirements for journal publication. Reconsider after major revision.

Author Response

Dear Reviewer,

Thank you for your letter and for the reviewers’ comments concerning our manuscript entitled “A Sustainable Iterative Product Design Method Based on Considering User Needs from Online Reviews”. Those comments are all valuable and very helpful for revising and improving our paper, as well as the important guiding significance to our research. We have studied the comments carefully and have made a correction which we hope meets with approval. Revised portions are marked in red on the paper. The major corrections in the paper and the responses to the reviewer's comments are as flowing:

Responds to the reviewer's comments:

Point 1:The motivation and purpose of the research fit for the theme goal of the journal, but whether the final research results are credible or not will have a key impact due to the characteristics of the product. To evaluate the use requirements, the relationship between the product life cycle and the design decision-making process, and drones may not necessarily be the best choice. Because the life cycle of the drone itself is not short as a phone, and online reviews are not intuitive, and it is even more difficult to convey the user experience in detail. Wearable devices, or products used in daily life may be more suitable as research and analysis targets, unless the author can provide more research data and information to explain the suitability of the research object. The author should add whether the product life cycle of drones is short and cite information such as the huge sales volume of the product to prove the suitability of the product category selection.

Response 1: We would like to thank you for your professional comments on our articles. As you may be concerned, we have added a quotation about the large sales volume of this type of UAV in section 4.1 cases, lines 301 to 302 of the article. Because of their versatility, drones have a large number of professional customers who take aerial photos while traveling. This kind of professional customer evaluation has a significant impact on how we approach product iteration decisions. Meanwhile, in China, we can see huge sales of DJI Mini 2 products in the JD shopping Malls. Using the following link, we can see the number of reviews over 100,000, and those reviews must be from buyers. As a result, we can see that sales are far greater than reviews. (https://item.jd.com/100042090900.html?cu=true&utm_source=kong&utm_medium=tuiguang&utm_campaign=t_1000124725_&utm_term=8988a8848fe849d28b8c54571edd8971)

Secondly, New UAVs are released much like mobile phones, about once a year. The imaging technology involved, battery capacity and so on also need to be updated with the development of The Times every year, UAV slowly began to belong to electronic fast-selling products. At the same time, there are also many vulnerable parts in UAVs, which will degrade the user experience, which also needs to be solved through continuous product update iteration.

All things considered, we chose drones as the case study. According to the reviewer's valuable suggestions, we will expand the research scope in the future and take wearable devices as the next research focus. Specific added references are as follows:

34. Stankovic, M.; Mirza, M. M.; Karabiyik, U., UAV Forensics: DJI Mini 2 Case Study. Drones 2021, 5, (2).

Point 2: In addition, there are a lot of numbers and calculation formulas in this article. In addition to presenting the numbers themselves, these econometric studies should also be supplemented with some corresponding visual charts to display the research results, such as bar charts and pie charts, or regression distribution curve maps. This article focuses on design research, the way and level of image presentation should also be paid attention to, and the quality of image expression should also be improved.

Response 2: We sincerely appreciate the valuable comments. We think this is an excellent suggestion. We have made changes, including converting the original Table 11 into a bar chart, which we have changed to Figure 3. This article does need to focus on the presentation of diagrams and tables. We would like to thank the reviewers again for their valuable comments. We have tried our best to improve the quality of pictures and text. In order to facilitate your browsing, the revised figure 3 is as follows:

Figure 3. Normalized mean weights of the six experts

Finally, we would like to express our sincere thanks to the reviewers for their valuable comments, which significantly improved the quality of our papers.

Thank you and best regards.

Yours sincerely,

Si Fu

Round 2

Reviewer 4 Report

After the author's response and the revision of the article, the content of the article has been greatly improved, and the added information on the purchase of drones does have issues that are more in line with the review opinions, such as the frequency of use of mobile phones. However, the suggestions for improvement of charts still need to be strengthened. It is not just about changing data into graphic charts, but also  to enhance visual effects and image quality, due to design field papers, it must have a certain degree of image quality. If improved graphics effects, I would recommend it to be published in this journal.

Author Response

We have prepared our response to the reviewers in a PDF document, which includes numerous figures and tables. We sincerely appreciate the reviewers' efforts in enhancing the quality of our paper. Thank you.
